# License to Kill: When iNKT Cells Are Granted the Use of Lethal Cytotoxicity

**DOI:** 10.3390/ijms21113909

**Published:** 2020-05-30

**Authors:** Angélica Díaz-Basabe, Francesco Strati, Federica Facciotti

**Affiliations:** 1Department of Experimental Oncology, IEO European Institute of Oncology IRCCS, 20139 Milan, Italy; angelicajulieth.diazbasabe@ieo.it (A.D.-B.); francesco.strati@ieo.it (F.S.); 2Department of Oncology and Hemato-Oncology, Università degli Studi di Milano, 20135 Milan, Italy

**Keywords:** iNKT, cytotoxicity, cancer, infections, CD1d

## Abstract

Invariant Natural Killer T (iNKT) cells are a non-conventional, innate-like, T cell population that recognize lipid antigens presented by the cluster of differentiation (CD)1d molecule. Although iNKT cells are mostly known for mediating several immune responses due to their massive and diverse cytokine release, these cells also work as effectors in various contexts thanks to their cytotoxic potential. In this Review, we focused on iNKT cell cytotoxicity; we provide an overview of iNKT cell subsets, their activation cues, the mechanisms of iNKT cell cytotoxicity, the specific roles and outcomes of this activity in various contexts, and how iNKT killing functions are currently activated in cancer immunotherapies. Finally, we discuss the future perspectives for the better understanding and potential uses of iNKT cell killing functions in tumor immunosurveillance.

Invariant natural killer T (iNKT) cells are a non-conventional T cell population co-expressing natural killer (NK)-lineage receptors and a semi-invariantly re-arranged T cell receptor (TCR) [1]. The TCR of these cells is usually composed by Vα24-Jα18 (human) or Vα14-Jα18 (mouse) α chains paired with a Vβ11 (human) Vβ8, -7, or -2 (mouse) β chain [1,2]. iNKT cells recognize microbial and self-lipid antigens presented by CD1d, a non-polymorphic, MHC class I-like molecule [3]. In mice, iNKT cells are mainly located in the liver and spleen (making up of 40% and 2% of the total T cell population); in the intestine they represent approximately 1% of total lymphocytes [3,4]. Less is known about iNKT cell distribution in humans, but they appear to be less abundant [1,4]. Due to the co-expression of NK receptors and a T cell receptor, as well as various cytokine receptors, iNKT cells respond very rapidly to TCR and/or cytokine signals with immediate production of cytokines participating to innate immunity and fostering adaptive immune responses [5].Because of their developmental pathway, that includes positive selection by thymocytes, iNKT cells are functionally mature even before they exit the thymus and promptly possess an antigen-experienced phenotype [6]. iNKT cells are tissue homing and tissue resident participating and, in many cases, mediating various immune responses in peripheral organs, from tissue homeostasis to microbial infections and tumor immunosurveillance, where they can exert either a protective or a pathogenic role depending on the context and the tissue involved [4]. In this review, we focus on how iNKT cell cytotoxicity plays a crucial role from the defense and clearance of cancer cells to infectious diseases. 

## 1. iNKT Cell Activation

The adaptive activation of iNKT cells is based on the recognition of CD1d-presented lipid antigens by the TCR (Figure 1). Exogenous antigens (Table 1) can enter the antigen-presenting cell via the mannose receptor or via endocytosis. All iNKT cells recognize and are potently activated by α-galactosylceramide (α-GalCer), a bacteria-derived glycolipid originally discovered in marine sponges [1]. Since then, several microbial antigens have been discovered both in pathogen and commensal microbes [7,8,9,10,11,12,13,14]. On the other hand, it has been demonstrated that the iNKT cell activation by self-antigens is important during thymic selection [1,15,16], infections [17,18,19,20], and autoimmune diseases [21]. Potential endogenous iNKT cell ligands include phosphatidylinositol, phosphatidylcholine, cardiolipin, sphingomyelin, lysophospholipids, gangliosides, and other glycosphingolipids, as they have been shown to bind CD1d [22].

Innate activation of iNKT cells, like natural killer cells, can be elicited through the balance between stimulatory and inhibitory signals via NKR (natural killer receptors) and KIR (killer-cell immunoglobulin-like receptor) [5,23]. Through the activation of some NKR, like NKG2D, iNKT cells can recognize stressed cells by the interaction with MHC-like molecules like MICA and MICB [5,24,25]. On the contrary, some NKRs and many KIRs produce inhibitory signals when they interact with classical HLA molecules [5,26]. Moreover, antigen-presenting cells, when stimulated with TLR ligands like LPS (lipopolysaccharide), secrete cytokines such as interleukins 12 and 18 that can activate iNKT cells in a CD1d-independent manner (Figure 1) [1,2,5,27,28]. 

## 2. iNKT Cell Subsets

iNKT cells are characterized by a massive and fast cytokine release shortly after activation. Human and murine iNKT cells can be classified into different functional subsets according to their cytokine secretion profile upon activation, in a way similar to the T helper classification [5,29]. NKT1 cells are similar to the T helper 1 (Th1) cells as they express the transcription factor T-bet and they secrete the typical Th1 cytokines, like IFN gamma and TNF, as well as cytotoxic molecules; NKT2 cells secrete IL-4 and IL-13; NKT17 cells are characterized by RORγt expression and IL-17A, IL-21, and IL-22 secretion [3,4]. Furthermore, an NKT10 subset has been identified in adipose tissue and intestinal polyps; these cells produce the anti-inflammatory cytokine IL-10, but, contrary to conventional Tregs, NKT10 cells do not express the transcription factor FOXP3, even if iNKT cells can express it after stimulation with TGF-β [4,29,30]. Although all subsets differentiate in the thymus, several lines of evidence have shown that further post-thymic differentiation takes place both in mice and humans [4].

After thymic development, iNKT cells can also be classified according to the expression of CD4 and CD8. In mice, iNKT cells can only be CD4+ or CD4-CD8- [2]. In humans, iNKT cells can be divided into CD4 CD8 double negative (DN), CD4 positive, or CD8 positive [2,4]. Circulating DN and CD8+ populations in humans are commonly associated with IFNγ secretion and cytotoxic activity similar to the NKT1 subset [4]. 

## 3. Immune Cell-Mediated Cytotoxicity Mechanisms

Only a few subsets of immune cells are capable of exerting cytotoxic functions. In the case of iNKT cells, this is a prerogative of the NKT1 subset, since this is the only population that expresses cytotoxic molecules [31]. In general, this NKT1 subset exert its cytotoxic activity through two main mechanisms: the death receptor pathway (or extrinsic apoptotic pathway), and the cytotoxic granule release (Figure 2) [32].

Members of the TNF family of ligands, Fas ligand (CD178, CD95L or FasL), TNF-related apoptosis ligand (TRAIL or Apo2 ligand), and TNFα are expressed by all cytotoxic populations, including iNKT cells [33,34,35,36]. Binding of these ligands to their respective receptors, Fas (CD95), death receptor 5 (DR5), and TNF receptor 1 (TNFR1), triggers the extrinsic apoptotic pathway [37]. Death ligands are usually bound to the cell membrane or packaged within cytotoxic granules [35]. Ligand-receptor interactions cause conformational changes that trigger the assembly of the DISC (death-inducing signaling complex), composed by the adaptor FAAD (Fas-associated DEATH domain protein, for Fas and DR5) or TRADD (TNFR1-associated DEATH domain protein), and pro-caspase 8. DISC formation leads to caspase 8 cleavage and activation, then caspase 8 cleaves and activates effector caspases 3 and 7, amplifying the death signal that leads to the apoptosis of the cell [37].

Perforin and granzymes (especially B and A) are the major components of cytotoxic granules present in all cytotoxic populations. Cytotoxic granules can be stored in the cytosol, as for NK cells, or produced only upon activation, as for conventional T cells [35,38,39]. This mechanism, as well as the death receptor pathway, requires cell-to-cell contact in the immunological synapse. Once the contact is established, the granules polarize towards the immunological synapse, and granzymes, perforin, and other components are released towards target cells via exocytosis. Perforin forms pores in the membranes of target cells to deliver granzymes and to induce lysis [35,40,41,42]. Five granzymes (A, B, H, K, and M) have been identified in humans, being granzyme B the most relevant in the induction of cell death [39]. Granzyme B is the most powerful and the fastest pro-apoptotic granzyme, as very low concentrations are sufficient to induce apoptosis; its efficiency is due to the fact that it can provoke cell death either in a caspase-dependent manner through pro-caspase 3, 7 and 8 cleavage, or caspase-independent manner through BH3-only protein BID, ICAD (inhibitor of caspase-activated DNase), poly(ADP ribose) polymerase (PARP), and lamin cleavage [39,43,44]. Granzyme A, on the other hand, induces apoptosis in a slower, caspase-independent manner, and it is poorly cytotoxic in humans [45,46].

A lesser known molecule, granulysin, is also stored in the in the granules of human iNKT cells, as in other immune cytotoxic populations [47,48]. It belongs to the family of the saposin-like proteins and exists in two isoforms: the 9 kDa isoform, found in cytotoxic granules together with granzymes and perforin, is endowed of antimicrobial and cytotoxic properties; instead, the 15 kDa isoform is involved in other immune functions, such as maturation of APCs and immune cell migration [48]. In general, granulysin has been mainly correlated with direct killing of pathogens because of its potent antimicrobial activity [47,48,49]. More specifically, the cytotoxic activity of granulysin mainly relies on its capacity of forming pores in the membranes of pathogens and, as it has been recently discovered, tumor cells; this capacity alters the membrane permeability of the cell inducing lysis [48,49]. Furthermore, similarly to perforin, granulysin mediates its cytotoxic activity by delivering granzymes which induce apoptosis in transformed cells and substantial oxidative damage in bacteria [48,49]. 

## 4. iNKT Cell Cytotoxicity in Response to Infections

It is widely known that iNKT cells and microorganisms are in constant crosstalk. This is achieved thanks to the capacity of iNKT cells to recognize microbial and microbe-induced self-antigens, as well as being activated by innate and cytokine signals, such as IL-12 and TLRs [50,51,52,53,54]. Given this, iNKT cells are important in the control of homeostasis in many mucosal tissues, like the intestine and lung, but also in the response against pathogens [9,50,51,52,53,54,55,56]. More specifically, iNKT cells participate in the response against infectious agents mainly by mediating the activation of other immune cell types through the release of interferon (IFN) gamma [9,50,51,52,53,54,55]. However, direct killing seems to play an important role as well. Pathogen clearance by iNKT cell cytotoxicity has been found to be relevant in *Leishmania infantum*, *Mycobacterium tuberculosis, Brucella suis,* Epstein–Barr (EBV) virus, and Hepatitis B (HBV) virus infections (Table 2). 

For instance, the iNKT cell role in the defense against *Leishmania* infection was established by several data [57,58,59], even if there is also evidence of a pathogenic role in visceral leishmaniasis [60]. The *Leishmania* species are intracellular protozoa that infect and survive inside phagocytes like neutrophils and macrophages [61]. It has been reported that iNKT cells are important in the control of *L. major* and *L. donovani* growth in vivo [57,62], and, more importantly, it has been found that they were capable of recognizing and directly eliminating *L. infantum*-infected dendritic cells due to the upregulation of CD1d [63]. Regarding the recognition of infected cells, it was previously reported that *L. donovani* synthesizes lipophosphoglycan, which was shown to activate a subset of hepatic iNKT cells when bound to CD1d [57]. The same or similar antigens could be present on other *Leishmania* species as well, but more studies must be performed on this matter. 

*Mycobacterium tuberculosis* is particularly successful for its ability to “hide” pathogen-associated molecular patterns (PAMPs) thanks to the composition of its lipid-enriched membrane, and for invading macrophages and dendritic cells [64]. Nonetheless, several data have shown that iNKT cells are capable of arresting *M. tuberculosis* growth [11,65,66,67]. In one of these studies, Gansert et al. showed that infected monocyte-derived cells were targeted and eliminated by iNKT cells in a CD1d-dependent manner through granulysin expression [67]. Moreover, it was later discovered that *M. tuberculosis*-derived phosphatidylinositol mannoside (PIM) is an antigen for iNKT cells that induces IFN-ɣ and TNF-ɑ secretion and lysis of CD1d-expressing HeLa cells [11], further explaining the mechanisms of iNKT cell recognition and elimination of infected cells in this context. More recently, Walker and collaborators observed that, in human immunodeficiency virus (HIV)-associated tuberculosis, iNKT cells are skewed towards a cytotoxic phenotype, as shown by the expression of CD107a, probably due to the depletion of CD4+ iNKT cells found in HIV infection [68]. 

*Brucella* species are facultative intracellular pathogens that cause fever, arthritis and osteomyelitis [69]. Bessoles and colleagues demonstrated that CD4+ iNKT cells recognized *B. suis*-infected macrophages in a CD1d-dependent manner and eliminated them via Fas ligand upregulation [69]. Despite it was demonstrated that iNKT cells targeted *B. suis*-infected macrophages through their TCR, no information about *B. suis*-derived iNKT antigens nor induction of self-antigens upon infection is available. 

Viruses are probably the most popular intracellular parasites, and some lines of evidence have highlighted the role iNKT cells in viral infections. For example, some studies have reported the importance of iNKT cell-mediated immunity in Epstein-Barr virus infection [70,71,72]. The works from Xiao et al. and Yuling et al. showed that iNKT cell numbers, especially CD8+ iNKT cells, were significantly lower in patients with advanced EBV-associated malignancies like Hodgkin’s lymphoma and nasopharyngeal carcinoma compared to healthy individuals and subjects with latent infection. Moreover, they demonstrated that CD8+ iNKT cells suppressed EBV-associated tumorigenesis in vitro and in vivo by eliminating EBV-infected cancer cells [70,71]. Furthermore, EBV provoked the downregulation of CD1d on infected B cells, event that significantly impaired iNKT cell cytotoxicity; it has been also observed that induced-CD1d expression on lymphoblastoid cell lines stimulated iNKT cell activation and killing even in the absence of α-GalCer, suggesting the role of an endogenous antigen in the recognition of EBV-infected cells [72]. 

Other studies have unraveled the role of other regulatory molecules in the response of iNKT cells to viruses. In this regard, it has been reported that the presence of Tim-3, a negative regulatory immune checkpoint, on iNKT cells impaired their activation and cytotoxic activity against HBV infected cells. Tim-3 blockade, both in vitro and *in vivo,* significantly increased iNKT cell-mediated inhibition of HBV propagation through IFN-ɣ and TNF-ɑ production, as well as cytotoxic granule release, as reflected by the increase of CD107a expression [73]. 

Despite the positive role of iNKT cell cytotoxic activity in some infections, this function can also contribute to pathogenesis and disease severity in others. In particular, iNKT cells have a relevant, pathogenic role in infection-derived liver injury. For instance, some studies have shown the detrimental role of iNKT cells during Dengue virus infection, which might be in part due to the increase of Fas ligand expression, which correlates with hepatocyte cell death [74]. Besides, during Salmonella infection in mice, TLR2 signaling induced the overexpression of Fas ligand on hepatic iNKT cells, resulting in hepatocyte death and increased liver damage [75]. In another study, Chen et al. assessed the role of intestinal pathogenic bacteria, like Salmonella, on iNKT cell cytotoxicity during concanavalin A-induced hepatitis, showing that pathogenic bacteria enhanced iNKT cell cytotoxicity in the liver via iNKT-dendritic cell interactions [76]. 

Even if iNKT cell cytotoxicity is mainly directed towards infected cells, they are also able to directly kill cellular pathogens. For example, iNKT cells are one of the main lines of defense against Borrelia burgdorferi, etiologic agent of Lyme disease [77,78,79]. In fact, diacylglycerol, a lipid produced by *B. burgdoferi*, is an antigen recognized by iNKT cells [7]. Accordingly, iNKT cells represent a protective barrier against *B. burgdoferi* invasion to the joints thanks to their granzyme B-dependent bactericidal activity. This activity is limited to joint-resident iNKT cells, as neither splenic nor hepatic iNKT cells were able to eliminate *B. burgdoferi in vivo* and even in in vitro contact experiments [77]. Another example of iNKT-mediated bactericidal activity is *M. tuberculosis*. Here, as it occurred with infected cells, iNKT cells exerted their bactericidal activity through granulysin release, as it is well-known for altering mycobacterial membranes [67]. 

Altogether, these data demonstrate that iNKT cell cytotoxic activity can be induced by microorganisms, and this response can be both protective or contribute to infection severity.

## 5. iNKT Cell Cytotoxic Activity in Other Diseases 

As it occurs in some infections, iNKT cell cytotoxicity can contribute to pathogenesis in other diseases (Table 2). For instance, iNKT cell pathogenic role in atherosclerosis has been validated in various murine studies [29]. Atherosclerosis is caused by the accumulation of low-density lipoproteins in the artery walls, which in turn unleashes an inflammatory response that gives rise to atherosclerotic plaques [80]. Regarding iNKT cells, one of their main proatherogenic roles is apoptosis. Indeed, Li et al. found that iNKT cells promoted atherosclerosis by inducing apoptosis in a perforin and granzyme B-dependent fashion, and these apoptotic processes further increased necrosis and inflammation [81]. Interestingly, the generation of atherosclerotic lesions by iNKT cells was dependent on cytotoxic proteins but not on cytokines, as IFN-ɣ, IL-4 and IL-21-deficient iNKT cells still augmented atherosclerosis. Nonetheless, cytotoxicity is not confined to iNKT cells in this context, as other T cell populations seem to increase their killing potential as well [80]. 

iNKT cells have also a detrimental role in allergic asthma [82,83]. It has been observed that iNKT cells from allergic asthma (AA) patients expressed higher levels of the activating NK receptors NKp30 and NKp46, perforin, and granzyme B, and iNKT cell cytotoxic phenotype positively correlated with disease severity, as granzyme B expression was significantly increased in patients suffering from severe to moderate AA compared to healthy individuals [82]. Moreover, it was demonstrated that CD4+ iNKT cells selectively eliminated autologous Treg cells in vitro [82], important in allergic asthma resolution as their numbers negatively correlate with IgE levels, and expression of CD39 and CD73 on Tregs has a negative correlation with Th17 cells, important in allergic inflammation [84].

Immunological hepatic injury is one of the most studied models in which iNKT cell cytotoxicity plays a pathogenic role. We have previously mentioned how pathogens induce iNKT cell-mediated liver injury by inducing Fas Ligand upregulation [76]. Furthermore, it has been observed that NK receptors, as well as TCR signaling play an important role in iNKT cell killing activity in liver injury. In particular, Kawamura and collaborators showed that NKG2A, an inhibitory NKR, inhibits iNKT cell activation and cytotoxic potential both in concanavalin A- and α-GalCer-induced hepatic injury, demonstrating that NKG2A is a major regulator in TCR-dependent iNKT cell activation [26]. Nonetheless, it was also observed that perforin-mediated killing, but not the Fas/FasL mechanism, is potentiated on iNKT cells in NKG2A-deficient mice, which indicated that not only Fas ligand upregulation is involved in iNKT cell killing of hepatocytes, but also the granzyme/perforin mechanism seems to have a relevant role. 

It is now known that hypoxia plays an important role in modulating immune cell functions in various diseases, especially cancer and inflammation [85,86,87]. iNKT cells are not the exception, as they are sensitive to hypoxia-inducible factor activation, enhancing CD1d-dependent activation and cytokine production [88]. However, in terms of cytotoxicity, the activation of the hypoxia-inducible factor 2α (HIF-2α) on iNKT cells acts as a protective factor in renal ischemia/reperfusion injury (IRI) [89]. Specifically, it was observed that, in HIF-2α-KO iNKT cell transfer experiments, IRI was exacerbated due to a higher iNKT infiltration and Fas ligand upregulation, being the Fas/FasL pathway of particular importance in IRI pathogenesis [89]. This elucidates a new role of hypoxia on IRI, as it inhibits iNKT cell pathogenic activity by controlling Fas ligand expression and iNKT cell recruitment.

## 6. iNKT Cytotoxic Activity in Antitumor Immunity 

iNKT cells have been recently praised for their multiple functions during tumor immunosurveillance [5,90,91,92,93]. In particular, iNKT cells are involved in dendritic cell maturation and IL-12 production, activation of CD4+ T cells, CD8+ T cells, and B cells, and NK cell transactivation [5,27,94]. Besides, iNKT cells are able to modify the tumor microenvironment by controlling immunosuppressive populations like tumor-associated macrophages and myeloid-derived suppressor cells [5,27,95,96]. More importantly, many lines of evidence have demonstrated iNKT cell effector functions in antitumor immunity, as peripheral blood-derived human and murine iNKT cells, as well as murine hepatic and splenic iNKT cells directly kill multiple types of cancer cells both in vitro and in vivo [38,96,97,98]. 

Cytotoxic activities displayed by iNKT cells have been mostly demonstrated in blood cancers. iNKT cells are capable of targeting hematopoietic cells due to the fact that CD1d is preferentially expressed in these cells [91]. In fact, CD1d-TCR interactions are of particular importance in chronic lymphoblastic leukemia, myelomonocytic leukemias, T-cell lymphoma, and acute myeloid leukemia (AML) cell recognition [36,99,100,101,102,103,104]. Nonetheless, leukemia cells can be also targeted via NKG2D, an activating NK receptor, in a TCR-independent manner [25]. Regarding the mechanisms used by iNKT cells to eliminate tumor cells in these contexts, it has been shown that the granzyme/perforin pathway is necessary to kill cancer cells in myelomonocytic leukemia [36,101], and T-cell lymphoma [100], whereas TRAIL expression drives apoptosis in AML cells [103]. Therefore, iNKT cells do not limit the identification and elimination of blood cancer cells to a certain mechanism, as they can target malignant cells both by innate and adaptive mechanisms and kill cells both by cytotoxic molecules and death receptor pathways. 

Although the iNKT cytotoxic activity has been mostly studied in hematopoietic cancers (further reviewed in [91]), iNKT cell effector activities have been also observed in solid tumors. In fact, several studies have demonstrated that human and murine iNKT cells isolated from peripheral blood, as well as murine liver, thymus, and spleen cells can directly kill colon [91,104,105], melanoma [97,98,104], lung [38,91,104,106], osteosarcoma [107], glioma [108], prostate [91], and breast cancer [91,109] cells in vitro and in vivo. 

For instance, iNKT cell infiltration in colon tumors has been considered a good prognostic factor because of the observation by Tachibana et al. that infiltrating iNKT cells express an activated phenotype, characterized by CD69, Fas ligand, and granzyme B expression [110]. In other studies, researchers have assessed the potential of chemotherapeutic agents and immune adjuvants in sensitizing colon cancer cells to iNKT cell killing, obtaining promising results mainly through CD1d upregulation [91,105]. While chemotherapy induced iNKT cell cytotoxicity in vitro through the expression of TRAIL and Fas ligand [91], human colon cancer cells treated with thymosin α1 (an immune adjuvant) and α-GalCer were eliminated by peripheral blood iNKT cells through granzyme B and perforin release, and the same treatment also reduced tumor growth in NOD-SCID mice [105]. 

Melanoma, together with leukemias, has become one of the types of cancer in which immunotherapy has obtained more attention [111]. Although immune checkpoint blockade therapy has been proved to have success in melanoma patients [111], efforts have also been done to establish the role and therapeutic potential of iNKT cells. In particular, Kawano and collaborators [112] showed that the cytotoxic activity of iNKT cells was significantly activated by α-GalCer, and this activation inhibited liver metastases of B16 melanoma cells. However, it was also observed that the TCR was not used in the killing interactions between iNKT cells and cancer cells, indicating an NK-like mechanism of target cell recognition; moreover, cytotoxic activity was not impaired by Fas ligand blockade, suggesting the involvement of other killing mechanisms. On the other hand, Wingender et al. [97] observed that there is a positive correlation between α-GalCer-mediated protection and CD1d expression by B16 cells in mice, suggesting that TCR-CD1d interactions are still important in melanoma cell elimination in vivo. 

iNKT cells and CD1d expression could be a prognostic factor in lung cancer as well. It was previously shown by Konishi et al., and confirmed by Dockry et al. that iNKT cell numbers are lower in lung cancer patients [38,106], and CD1d expression in non-small cell lung cancer NSCLC was positively correlated with overall survival [106]. Regarding iNKT cell cytotoxicity in this context, some lines of evidence have shown that peripheral blood-derived iNKT cells can eliminate lung cancer cells in vitro [38,91,104,106]. As it occurred in other types of cancer, iNKT cells could target cancer cells both in a CD1-dependent and CD1d-independent manner [38,91,104,106]. In this regard, it was also demonstrated that epigenetic induction of CD1d expression enhanced iNKT cell killing [106]. Plus, it has been also observed that degranulation and perforin release are increased and fundamental for iNKT cell cytotoxicity in vitro [38,106]. 

Despite most of the research in iNKT cell-mediated killing activities against tumors have focused on cancer cell elimination, some studies have addressed the role of iNKT cell cytotoxicity in the control of the tumor microenvironment, especially of CD1d-positive, myeloid-derived cells. More specifically, tumor-associated macrophages (TAMs) seem to be sensitive to iNKT cell elimination in neuroblastoma and prostate cancer models. In particular, TAMs from neuroblastoma tumors are capable of cross-presenting neuroblastoma-cell glycosphingolipids, inducing a potent cytotoxic activity by infiltrating iNKT cells against these APCs [98]. On the other hand, in the TRAMP model of oncogene-induced prostate cancer, Cortesi and colleagues [113] demonstrated a complex TAM control by iNKT cells, as they only eliminated M2-like macrophages via Fas/Fas ligand binding, but supported M1-like (antitumor) macrophages via CD40 ligand. 

Hence, these results illustrate how innate and adaptive signals can unleash iNKT cell killing potential against tumors via soluble and membrane-bound cytotoxic proteins (Figure 3). We have previously mentioned that iNKT cells express NKG2D, which allows the recognition of stressed cells that express MICA, MICB, among others, and how this receptor drove iNKT cell cytotoxic activity both in a fully CD1d-independent manner or acting as TCR co-stimulator against leukemia cells [25,114,115,116]. Other activating innate receptors expressed by iNKT cells, such as NKp30, NKp46 and DNAM1, might also be involved in their response against tumors, as they are positively involved in natural killer and cytotoxic T cell cytotoxicity [25,82,117,118]. Cytokines can also activate iNKT cell killing activity without TCR stimulation, as it was shown for IL-18 plus IL-12 on Fas+ target cells [119]. While the innate signals received by iNKT cells can be explained by the presence of natural killer receptor-ligand interactions and cytokine activation, the lipid antigens specifically presented by tumor cells is a more intriguing topic. In this regard, it was demonstrated that iNKT cell cytotoxicity in vivo positively correlated with CD1d expression and antigen potency [97]. Some glycolipids, like gangliosides GD2, GD3, and GM2 are synthesized and presented in an altered manner in many cancers (90), and different cellular stresses commonly found in cancer cells, such as endoplasmic reticulum stress, enhance antigen-dependent iNKT activation [120,121]. However, tumor-specific antigens that activate iNKT cells are still largely unknown.

## 7. iNKT Cell-Based Cancer Immunotherapies

Due to the important roles of iNKT cells in cancer immunosurveillance, including the ability to kill cancer cells, iNKT cell-based immunotherapies are currently used in the fight against cancer. Most of the ongoing clinical trials on solid tumors take advantage of the strong activation given by α-GalCer through direct injection. In these, it was found an increase in cytokines typical of the NKT1 subset, such as TNF-α and IFN-ɣ, as well as GM-CSF [122]. However, iNKT cell anergy was increased in studies in which only α-GalCer was administered [90,94,122]. To overcome the limitations of soluble α-GalCer therapy, some trials have used α-GalCer-pulsed, autologous dendritic cell transfer as an alternative treatment, being well tolerated and inducing a positive response in some of them, measured by an increase in Th1 cytokines and an expansion in IFN-ɣ-producing iNKT cells [94,115,122,123]. Other efforts have been done in antigen research to identify stronger iNKT cell agonists or to modify α-GalCer. In this regard, several α-GalCer analogs were shown to elicit a strong Th1 response on iNKT cells [27,90]. Other strategies, like the modulation of metabolic pathways have also been explored for iNKT cell-based therapies. In relation to this, Fu and colleagues found that increasing iNKT cell lipid biosynthesis (mostly cholesterol synthesis) increased IFNγ production by intratumoral iNKT cells both in patients and mice [124].

More recently, adoptive cell therapies, mostly CAR-T therapy, have gained popularity especially in hematopoietic cancers, although several advances have also been done in solid tumor treatment [125,126,127,128]. iNKT cells are not the exception, as new lines of evidence have shown their potential both in preclinical and clinical studies [94,115]. CAR-T iNKT cells specific for GD2 ganglioside tested in preclinical studies for neuroblastoma and B-cell lymphoma have shown promising results, and patients have been recruited for phase I studies [27,94,115]. Moreover, studies using endogenous iNKT TCR have shown that these cells induced a strong increase in tumor cell death against various types of cancer [94]. 

## 8. Concluding Remarks

iNKT cells are an important component of innate and adaptive immunity also due to their cytotoxic functions. Although iNKT cell cytotoxicity is considered protective in responses against some infections and mostly in antitumor immunity, it also plays a negative role, enhancing pathogenesis in other immune-mediated diseases. Therefore, it is essential to decipher the signals by which iNKT cell killing functions are activated in each setting. This is particularly important in cancer research, as it is still unknown which lipid antigens and innate signals are specifically presented by tumor cells and are responsible for iNKT cell activation. This knowledge would help to understand how iNKT cell cytotoxic activity is unleashed against specific targets in order to exploit these cells in immunotherapies. Moreover, the assessment of iNKT cell triggering signals would be useful to find therapies that would arrest the iNKT cell pathogenic role in other malignancies.

## Figures and Tables

**Figure 1 ijms-21-03909-f001:**
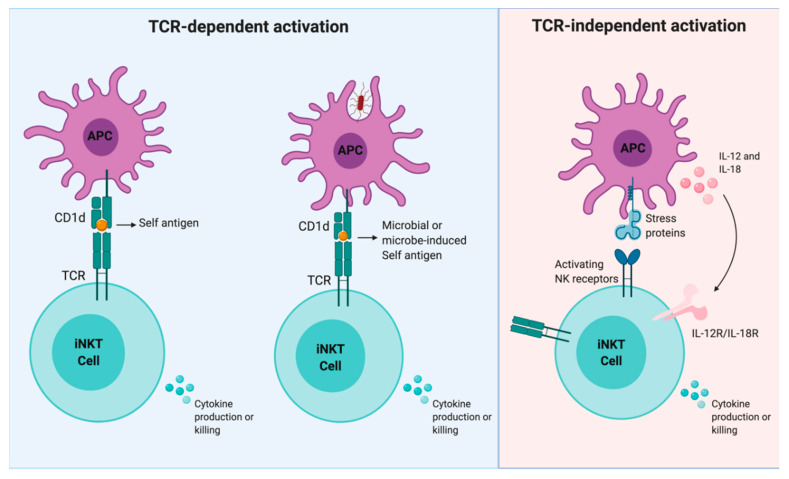
Mechanisms of iNKT cell activation.

**Figure 2 ijms-21-03909-f002:**
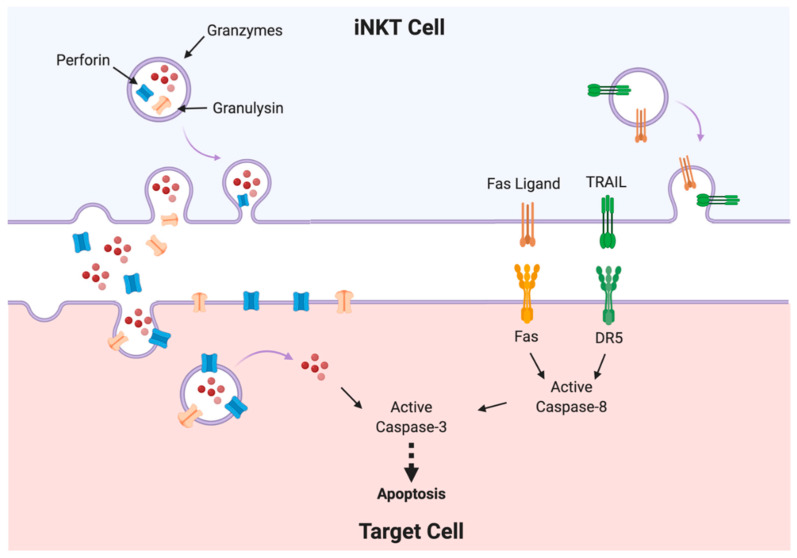
Cytotoxicity mechanisms used by iNKT cells.

**Figure 3 ijms-21-03909-f003:**
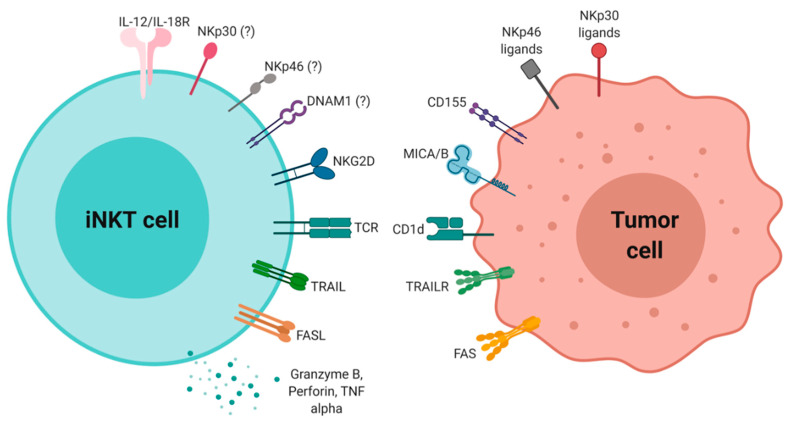
Activation signals and mechanisms of NKT cell cytotoxicity against tumors.

**Table 1 ijms-21-03909-t001:** Microbial antigens recognized by iNKT cells (adapted from [9]).

Microorganism	Pathogenicity	Antigen
*Arthrobacter*	Commensal, opportunist	M-AcM-MAG
*Aspergillus fumigatus* and *Aspergillus niger*	Opportunists	Asperamide B
*Bacteroides fragilis*	Commensal, opportunist	α-GalCer (Bf)
*Bacteroides vulgatus*	Commensal, opportunist	α-GalCer
*Borrelia burgdorferi*	Pathogen	BbGL-II (1,2-di-O-acyl- 3-O-a Dgalactopyranosyl-sn-glycerol, 6)
*Candida albicans*	Commensal, opportunist	ChAcMan
*Ehrlichia muris*	Pathogen in rodents only	Not defined
*Entamoeba histolytica*	Opportunist	Lipopeptidophosphoglycan (EhLPPG)
*Helicobacter pylori*	Commensal, opportunist	Cholestoryl-a-glucosides, especially monoacyl a-CPG
*Lactobacillus casei*	Commensal	Glc-DAG
*Leishmania donovani*	Opportunist	Lipophosphoglycan (LPG)
*Mycobacterium tuberculosis*	Pathogen	Phosphatidylinositol mannoside (PIM)
*Prevotella copri*	Commensal	α-GalCer
*Rothia dentocariosa*	Commensal, opportunist	M-AcM-MAG
*Saccharopolyspora*	Environmental, opportunist	M-AcM-MAG
*Sphingomonas paucimobilis*	Commensal, opportunist	a-glucuronosyl ceramide (GSL-1/ aGlcUCer)
*Sphingomonas yanoikuyae*	Environmental, commensal, opportunist	a-galacturonosyl-ceramides
*Sphingomonas wittichi*	No pathogenicity reported	a-galacturonosyl-ceramides
*Streptococcus pneumoniae* and Group B *Streptococcus*	Commensal, opportunists	SPN-Glc-DAG, SPN-Gal-Glc-DAG

**Table 2 ijms-21-03909-t002:** Role of iNKT cell cytotoxicity in disease.

Disease	Role of iNKT Cell Cytotoxicity	Killing Mechanism	References
*Leishmania infantum* infection	Protective	Not addressed	[63]
*Brucella suis* infection	Protective	Fas ligand upregulation	[69]
Epstein-Barr virus infection	Protective	Infected cell killing by IFN gamma and TNF alpha production	[70,71,72]
*Borrelia burgdoferi* infection	Protective	Bacteria death by Granzyme B release	[77]
*Mycobacterium tuberculosis* infection	Protective	Infected cell and bacteria elimination by granulysin release	[67]
Hepatitis B virus infection	Protective	Elimination of infected cells by IFN gamma, TNF alpha production and cytotoxic granule release	[73]
Atherosclerosis	Pathogenic	Granzyme B and perforin release	[81]
Allergic asthma	Pathogenic	Increase in granzyme B and perforin. Killing of Tregs in vitro	[82]
Liver injury	Pathogenic	Hepatocyte cell death by Fas ligand upregulation, perforin and granzyme B release	[26,74,75,76]
Renal ischemia/reperfusion injury	Pathogenic	Fas ligand upregulation	[89]

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
