# Peer review of "License to Kill: When iNKT Cells Are Granted the Use of Lethal Cytotoxicity"

_ijms, 2020, doi:10.3390/ijms21113909_

Round 1
Reviewer 1 Report
The manuscript deals with a very interesting and important topic. The review aims at summarizing current knowledge about NKT cell cytotoxicity in physiological and pathological conditions. The work is well and comprehensive written, sections are logically organized and the subject is appropriately discussed in detail. Only some additional illustrations and data could increase further the quality of the paper. Therefore, my suggestions:
- a summarizing table about all possible microbial antigens capable of being recognized by iNKT cells
- possible activation mechanisms of iNKT(CD1d dependent and independent) should be demonstrated in a figure similar to Figure 1
- the section about iNKT cell subsets should be completed with other grouping aspects, like CD4+, CD8+ iNKT cells, as mentioned by EBV infection
Author Response
Response to Reviewer 1
We thank the reviewer for the comments that enabled us to improve the quality of our manuscript.
- According to the reviewer’ suggestion we added a table including all the microbial antigens recognized by iNKT cells from Line 60 to Line 62.
- We thank the reviewer for the suggestion. Accordingly, we added a Figure of the mechanisms to activate iNKT cells in Lines 63-64
- We also added a paragraph of iNKT cell classification according to the CD4/CD8 expression from Line 78 to 82
Reviewer 2 Report
The manuscript presented by Angélica Díaz-Basabe, Francesco Strati and Federica Facciotti reviews the cytotoxic capacity of a subpopulation of lymphocytes with an semi-invariantly re-arranged T cell receptor, iNKTs: the iNKT activation, differents ubsets, the mechanisms of cytotoxicity described for these cells, including apoptosis and release of cytotoxic granules (a scheme similar to figure 1 would be interesting), and reviews the cytotoxic capacity described for these cells in response to intracellular pathogen infections and a further description of the role described for iNKTs in antitumor immunity and their potential as antitumor agents (use of alpha galactosyl ceramide and CAR-T iNKT cells).
Authors should review the most recent 2020 references. For example:
Impaired lipid biosynthesis hinders anti-tumor efficacy of intratumoral iNKT cells.
Nat Commun. 2020 Jan 23;11(1):438. doi: 10.1038/s41467-020-14332-x.
Invariant Natural Killer T-cell Dynamics in Human Immunodeficiency Virus-associated Tuberculosis.
Clin Infect Dis. 2020 Apr 15;70(9):1865-1874. doi: 10.1093/cid/ciz501.
Author Response
Response to Reviewer 2
We thank the reviewer for the insightful papers that we added to the manuscript improving our manuscript
- According to the reviewer’s suggestion, we added a scheme of the cytotoxicity mechanisms used by iNKT cells in Lines 127-128.
- According to the reviewer’s comment, we added the suggested references in Lines 158-161, and 353-356. We also included another recent reference in Line 337